# Model-tuning Via Prompts
# Makes NLP Models Adversarially Robust

**Mrigank Raman**[*]
Carnegie Mellon University
mrigankr@cmu.edu

**Pratyush Maini**[*]
Carnegie Mellon University
pratyushmaini@cmu.edu

**Zico Kolter**
Carnegie Mellon University
zkolter@cs.cmu.edu

**Zachary C. Lipton**
Carnegie Mellon University
zlipton@cmu.edu

**Danish Pruthi**
Indian Institute of Science (IISc)
danishp@iisc.ac.in

## Abstract

In recent years, NLP practitioners have converged on the following practice: (i) import an off-the-shelf pretrained (masked) language model; (ii) append a multilayer perceptron atop the CLS token's hidden representation (with randomly initialized weights); and (iii) fine-tune the entire model on a downstream task (`MLP-FT`). This procedure has produced massive gains on standard NLP benchmarks, but these models remain brittle, even to mild adversarial perturbations. In this work, we demonstrate surprising gains in adversarial robustness enjoyed by Model-tuning Via Prompts (`MVP`), an alternative method of adapting to downstream tasks. Rather than appending an MLP head to make output prediction, `MVP` appends a prompt template to the input, and makes prediction via text infilling/completion. Across 5 NLP datasets, 4 adversarial attacks, and 3 different models, `MVP` improves performance against adversarial substitutions by an average of $8\%$ over standard methods and even outperforms adversarial training-based state-of-art defenses by $3.5\%$. By combining `MVP` with adversarial training, we achieve further improvements in adversarial robustness while maintaining performance on unperturbed examples. Finally, we conduct ablations to investigate the mechanism underlying these gains. Notably, we find that the main causes of vulnerability of `MLP-FT` can be attributed to the misalignment between pre-training and fine-tuning tasks, and the randomly initialized MLP parameters.[1]

## 1 Introduction

Pre-trained NLP models (Devlin et al., 2019; Liu et al., 2019) are typically adapted to downstream tasks by (i) appending a randomly initialized multilayer perceptron to their topmost representation layer; and then (ii) fine-tuning the resulting model on downstream data (`MLP-FT`). More recently, work on large language models has demonstrated comparable performance without fine-tuning, by just prompting the model with a prefix containing several examples of inputs and corresponding target values (Brown et al., 2020). More broadly, prompting approaches recast classification problems as sequence completion (or mask infilling) tasks by embedding the example of interest into a prompt template. The model's output is then mapped to a set of candidate answers to make the final prediction. Prompting has emerged as an effective strategy for large-scale language models (Lester et al., 2021), and its utility has also been demonstrated for masked language models (Gao et al., 2021).

While fine-tuned models perform well on in-distribution data, a growing body of work demonstrates that they remain brittle to adversarial perturbations (Jin et al., 2020; Li et al., 2020; Morris et al., 2020a). Even small changes in the input text, such as replacement with synonyms (Ebrahimi et al., 2018b), and adversarial misspellings (Ebrahimi et al., 2018a; Pruthi et al., 2019) drastically degrade the accuracy of text classification models. While prompting has become a popular approach for adapting pretrained models to downstream data, little work has considered interactions between adaptation strategies and adversarial robustness.

In this work, **first**, we demonstrate surprising benefits of Model-tuning Via Prompts (`MVP`) in terms of robustness to adversarial substitutions, as compared to the standard approach of fine-tuning models with an MLP head (`MLP-FT`). Notably, `MVP`, which does not utilize any sort of adversarial training or prompt optimization/engineering already yields higher adversarial robustness compared to the state-of-the-art methods utilizing adversarial training by an average of $3.5\%$ across five datasets (classification, boolean question answering, and paraphrase detection), 3 models (BERT, RoBERTa, and GPT-2) and four attacks (word and character-level substitutions) (§5). Moreover, we find that combining

---

[*] Equal contribution.
[1] Code is available at https://github.com/acmi-lab/mvp.

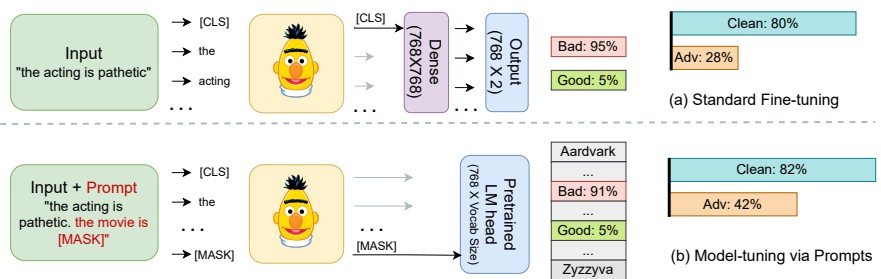

Figure 1: An illustration of (a) Standard Finetuning, and (b) Model-tuning via Prompts. The adjoining accuracy metrics correspond to a RoBERTa model trained on the BoolQ dataset.

MVP with single-step adversarial training can further boost adversarial robustness, resulting in combined robustness gains of more than 10% over the baselines. This happens without any loss in accuracy on unperturbed inputs, indicating how the objective of adversarial training couples well with MVP.

So far, prior works have not explored the idea of fine-tuning all the parameters of a model via prompts (we call this setup full-model full-data fine-tuning). We only see instances of (i) fine-tuning the full model via prompts in a few-shot setting (Gao et al., 2021), or (ii) fine-tuning additional tunable parameters using prompts on top of a frozen model by utilizing the complete training set (Li and Liang, 2021). We believe the idea of full-model full-data fine-tuning via prompts has not been used until now because clean accuracy improvements for MVP over MLP-FT are negligible, and the robustness advantages of MVP were previously undiscovered.

**Second**, we show that MVP as a method for classification is more (i) sample efficient, and (ii) has higher effective robustness than MLP-FT (§5.1). That is, MVP requires fewer training samples to achieve the same clean accuracy; and for any given clean accuracy, the robust accuracy of MVP is higher than MLP-FT. Through ablation studies (§5.3), we find that (i) adding multiple prompt templates makes it harder to fool the model; and (ii) having multiple candidate answers has a small but positive impact on the robustness.

**Third**, to explain our observations, we test a set of hypotheses (§6), including (i) *random parameter vulnerability*—is adding a randomly initialized linear head the source of adversarial vulnerability for MLP-FT?; (ii) *pretraining task alignment*—can the gains in robustness be attributed to the alignment between the fine-tuning and pretaining tasks in MVP?; and (iii) *semantically similar candidates*—are predictions by MVP more robust because the candidate answer is semantically similar to the class label?

Through experiments designed to test these hypotheses, we find that (i) in the absence of pretraining, MVP and MLP-FT have similar robustness performance, supporting the hypothesis of pretraining task alignment; (ii) adding extra uninitialized parameters to MVP leads to a sharp drop in robustness, whereas removing the dense (768,768) randomly initialized weight matrix from MLP-FT improves the robustness of the model significantly; (iii) even random candidate answers such as 'jack', and 'jill' result in similar robustness gains, suggesting that when fine-tuning through prompts, the choice of candidate answers is inconsequential (in contrast, this choice is known to be crucial for few-shot classification).

**Fourth**, we perform a user study (§7) to assess the validity of adversarial examples. We find that human annotators were 23% more likely to find adversarial examples to have been perturbed as opposed to clean examples. Moreover, humans achieved 11% lower accuracy on adversarial examples as compared to clean examples with average confidence on the label of perturbed examples being 15% lower. This highlights that a large fraction of adversarial examples are already detected by humans, and often change the true label of the input, signifying that MVP is more robust than crude statistics discussed in §5. Future work will benefit from developing better evaluation strategies for the robustness of NLP models.

**Fifth**, going beyond adversarial robustness, we investigate the robustness gains of MVP over MLP-FT on out-of-distribution (OOD) tasks. We find that MVP improves robustness by 2% across 5 different OOD sentiment analysis tasks (§ 5.2).

In summary, we demonstrate that models tuned via prompts (MVP) are considerably more robust than the models adapted through the conventional approach of fine-tuning with an MLP head. Our findings suggest that practitioners adopt MVP as a means of fine-tuning, regardless of the training data size (few-shot or full data) and model capacity.

## 2 Related Work

**Adversarial Attacks and Defenses** Inspired by the brittleness of vision models to adversarial examples (Szegedy et al., 2013; Goodfellow et al., 2014), researchers have found similar vulnerabilities to also exist in language models (Alzantot et al., 2018; Belinkov and Bisk, 2018). Unlike vision, the goal in NLP is to develop (i) semantically viable substitutions or deletions (Ebrahimi et al., 2018b); (ii) character-level misspellings (Zhang et al., 2015b; Ebrahimi et al., 2018a; Pruthi et al., 2019); or (iii) imperceptible homoglyphs (Boucher et al., 2022).

The discovery of such adversarial examples span several tasks such as classification (Zhang et al., 2015b; Alzantot et al., 2018), NMT (Belinkov and Bisk, 2018), and question-answering (Jia and Liang, 2017), but they are restricted to small models such as LSTMs and RNNs. Among others, Jin et al. (2020); Li et al. (2020) show that despite producing massive gains on standard NLP benchmarks, BERT style pretrained models are susceptible to adversarial attacks when finetuned on downstream tasks. Subsequently, multiple works have attempted at developing fast and semantically meaningful attacks (Li et al., 2018) and scalable defenses (Wang and Bansal, 2018; Jia et al., 2019; Wang et al., 2021b; Si et al., 2021b; Zhu et al., 2020) for masked language models. Yang et al. (2022) leverage prompts to generate adversarial examples that they train their model on using MLP-FT. Despite these efforts, NLP models suffer a significant drop in robust accuracy, when compared to clean accuracy on the same task.

**Prompting NLP Models** Prompting gained traction from GPT-3 (Brown et al., 2020) where it was primarily used in the zero-shot and few-shot settings and required manual trials to increase performance. In the zero-shot setting, no labeled examples are provided to the model and the language model is kept frozen. The model needs to output its prediction using the prompt that is provided. Whereas, in the few-shot setting, a few task-specific labeled examples are also provided for the frozen model in addition to the prompt (also known as in-context learning) (Rubin et al., 2022; Levine et al., 2022). A lot of work has gone into improving the prompts that are used in the zero-shot and few-shot settings, including mining-based methods to automatically augment prompts (Jiang et al., 2020), gradient-based search (Shin et al., 2020), using generative language models (Gao et al., 2021) and others (Hu et al., 2022; Schick

and Schütze, 2021b,a). In the full data setting, previous works have explored prompting via prompt tuning (Liu et al., 2022; Li and Liang, 2021; Qin and Eisner, 2021; Lester et al., 2021) where the model is injected with additional tunable parameters. None of these works discuss the robustness advantages of prompting (especially in the adversarial context) when compared to standard fine-tuning approaches.

**Robust Fine-tuning and Adaptation** In the vision literature, prior works have also tried to use prompting to improve out-of-distribution robustness in the zero-shot and few-shot settings (Zhou et al., 2022a,b). Kumar et al. (2022) observed that fine-tuning worsens the out-of-distribution (OOD) performance of models due to the bias introduced via a randomly-initialized head on top of the CLIP model, and instead suggest a procedure (LPFT) that first fits the linear head and then finetunes the model. Later works have shown that this ID/OOD performance trade-off could be mitigated by averaging model weights between the original zero-shot and fine-tuned model (Wortsman et al., 2022) and/or by finetuning using an objective similar to that used for pretraining (Goyal et al., 2022). However, this work has been applied only to vision–language models, and secondly only deals with "natural" robustness evaluations rather than the adversarial manipulations we consider here.

## 3 Method

We consider the task of supervised text classification, where we have a dataset $\mathcal{S} = \{x^{(i)}, y^{(i)}\}^n$, with $x^{(i)} \in \mathcal{X}$ and $y^{(i)} \in \{1, ..., k\}$ for a $k$-class classification problem. We train a classifier $f$ to predict $y$ based on input $x$. We follow the terminology by Schick and Schütze (2021a). The input $(x)$ can be decomposed as a sequence of words $\{x_1, x_2, ..., x_l\}$, and the output $(y)$ is a positive integer, with each value corresponding to a particular class. The prompt template $(t)$ is the input string we append at the beginning or end of the input. For example, we may append the following template at the end of a movie review—"This movie is [MASK]". The candidate answers $(\mathcal{A})$ is a set of words corresponding to each class. For example, the positive sentiment class can have the following candidate answers—{great, good, amazing}.

**Adversarial Attacks** We are concerned with perturbations to the input $x$ that change the model prediction. In the case of adversarial attacks confined

to synonym substitutions, we confine the model to searching for $\hat{x}_i$ in the synonym set of every word $x_i$ in the input. Whereas, in the case of character level substitution, we consider substitutions of characters that compose each $x_i$ in the input.

## 3.1 Model-tuning Via Prompts (`MVP`)

We present the overall pipeline of `MVP` in Figure 1(b), and describe individual components below.

**Input Modification** Consider a prompt template $t = t_1, t_2, ... \texttt{[MASK]}, ... t_m$. For any input $x$, the prompt input ($x_t$) can be constructed by appending the template at the beginning or end of the input. The final output is based on the most likely substitution for the `[MASK]` token, as given by the language model. Typically, we use a set of prompt templates denoted by $\mathcal{T}$.

**Inference** For every class label, we have a set of candidate answers associated with it. During inference, we do the following: (i) for every class label, select the candidate corresponding to the largest logit value among its candidate set; (ii) take the mean of the logits corresponding to the selected candidates over all the templates to compute the final logit of the given class label; (iii) predict the class having the highest final logit.

## 3.2 `MVP` + Single-step Adv

Based on the Fast Gradient Sign Method (FGSM) by Goodfellow et al. (2014), we perform single-step adversarial training. Note that the input tokens are discrete vectors, and hence it is not possible to perturb the inputs directly. Instead, we pass the inputs through the embedding layer of the model and then perform adversarial perturbations in the embedding space. We do not perturb the embeddings corresponding to the prompt tokens. We find that performing single-step perturbations with the $\ell_2$ constraint leads to more stable training than in the $\ell_\infty$ norm ball, and use the same for all our experiments. Similar (but not equivalent) approaches have also been studied in literature (Si et al., 2021a).

## 4 Experimental Setup

**Datasets and Models** We perform our experiments on five different datasets—AG News (Zhang et al., 2015b) (4-class topic classification), SST-2 (Socher et al., 2013) (binary sentiment classification), BoolQ (Clark et al., 2019) (boolean question answering), DBPedia14 (Zhang et al., 2015a)

(14-class topic classification), and MRPC (Dolan and Brockett, 2005) (paraphrase detection). Results on DBPedia14 and MRPC are presented in Appendix C.1. All models are trained with the RoBERTa-Base (Zhuang et al., 2021) backbone. Experiments on GPT-2 and BERT-Base (Devlin et al., 2019) are included in Appendix C. Detailed information about training and attack hyperparameters is provided in Appendix E.

**Attack Strategies** We perturb the inputs using the TextAttack library (Morris et al., 2020b). In particular, we use 1 character-level attack and 3 word-level attacks. Word-level attacks include the TextFooler (Jin et al., 2020), TextBugger (Li et al., 2018) that replace words with neighboring words based on counterfitted GloVe embeddings and BertAttack (Li et al., 2020) that uses BERT to replace keywords with synonyms.[2] For character-level attack, we use adversarial misspellings (Pruthi et al., 2019). More details are in Appendix B.2.

**Baseline Methods** We now describe the terminologies used to denote training schemes corresponding to various fine-tuning strategies. `MLP-FT` is the "base" model for classification via standard non-adversarial training, and is utilized by all the baselines. Given a pre-trained model, we perform downstream fine-tuning by adding an MLP layer to the output corresponding to `[CLS]` token as illustrated in Figure 1(a). This hidden representation is of size $768 \times 1$. In the case of the BERT model, there is a single dense layer of dimension $768 \times 2$, whereas in the case of RoBERTa model, we have a two-layer MLP that is used to make the final prediction. `MLP-FT` + Adv is is identical to the method used for adversarial training in Section 3.2, wherein we perform adversarial perturbations in the embedding space of the `MLP-FT` model, rather than `MVP`. To compare with state-of-art adversarial training-based defenses we consider FreeLB++ (Li et al., 2021) (free large batch adversarial training using projected gradient descent), InfoBERT (Wang et al., 2021a) (information bottleneck regularizer to suppress noisy information), and AMDA (Si et al., 2021b) (adversarial and mixup data augmentation for creating new training examples via interpolation). We provide complete details pertaining to each baseline method in Appendix B.1.

---

[2] In line with previous benchmark (Li et al., 2021) we only use the word-substitution transformation in TextBugger.

| | MLP-FT | MLP-FT + Adv | Free LB++ | MADA | InfoBert | MVP | MVP + Adv |
|---|---|---|---|---|---|---|---|
| | | | | **SST2** | | | |
| Clean Acc | 93.6 ±0.4 | 93.6 ±0.6 | 94.0 ±0.1 | 93.8 ±0.4 | 94.0 ±0.4 | 93.9 ±0.7 | 93.8 ±0.1 |
| TextFooler | 40.2 ±0.9 | 44.0 ±1.2 | 43.4 ±1.0 | 41.8 ±0.5 | 43.6 ±0.5 | 46.9 ±0.5 | **53.8 ±0.7** |
| TextBugger | 65.4 ±0.3 | 68.5 ±1.5 | 67.2 ±0.6 | 66.1 ±0.2 | 66.6 ±1.8 | 69.8 ±0.5 | **71.7 ±0.8** |
| BertAttack | 70.3 ±0.9 | 74.3 ±0.8 | 76.2 ±0.6 | 74.2 ±0.2 | 76.1 ±0.6 | 78.1 ±0.9 | **81.7 ±0.7** |
| Misspellings | 45.2 ±1.1 | 49.3 ±0.3 | 50.4 ±1.1 | 45.4 ±0.4 | 47.1 ±0.4 | 50.5 ±0.7 | **54.9 ±1.3** |
| | | | | **AG News** | | | |
| Clean Acc | 94.5 ±0.4 | 94.4 ±0.6 | 94.4 ±0.7 | 94.1 ±0.6 | 94.5 ±0.9 | 94.3 ±0.2 | 94.4 ±0.8 |
| TextFooler | 42.9 ±0.7 | 47.7 ±0.5 | 46.9 ±1.6 | 44.3 ±1.4 | 48.0 ±2.2 | 51.5 ±2.1 | **62.7 ±2.4** |
| TextBugger | 61.8 ±0.3 | 65.6 ±0.8 | 65.5 ±1.0 | 62.9 ±0.5 | 65.6 ±1.2 | 68.7 ±0.7 | **75.3 ±1.6** |
| BertAttack | 79.1 ±1.3 | 81.1 ±1.0 | 81.4 ±0.9 | 80.4 ±0.2 | 82.4 ±1.2 | 85.3 ±1.3 | **88.2 ±0.9** |
| Misspellings | 76.8 ±1.3 | 78.6 ±0.8 | 80.1 ±1.3 | 77.1 ±0.4 | 80.4 ±1.4 | 82.7 ±0.7 | **86.6 ±0.6** |
| | | | | **BoolQ** | | | |
| Clean Acc | 80.6 ±1.5 | 78.9 ±1.2 | 80.6 ±0.4 | 79.2 ±0.9 | 81.5 ±0.7 | 82.0 ±0.6 | 81.1 ±0.6 |
| TextFooler | 28.2 ±1.7 | 39.0 ±0.7 | 37.2 ±1.4 | 32.0 ±0.3 | 38.0 ±1.3 | 42.9 ±0.5 | **52.2 ±1.6** |
| TextBugger | 38.3 ±1.0 | 44.4 ±1.2 | 43.2 ±1.0 | 41.1 ±0.2 | 42.4 ±1.5 | 46.8 ±0.9 | **56.7 ±1.2** |
| BertAttack | 48.1 ±0.7 | 57.6 ±1.3 | 57.3 ±1.5 | 55.2 ±0.4 | 57.4 ±1.0 | 61.5 ±1.2 | **69.4 ±1.5** |
| Misspellings | 42.9 ±1.0 | 47.4 ±1.1 | 46.6 ±1.1 | 45.2 ±0.3 | 47.3 ±1.2 | 51.6 ±0.8 | **59.7 ±1.0** |

Table 1: **Adversarial Robustness:** Performance of RoBERTa-base model on 3 different datasets averaged over 3 different seeds on a fixed test set of size 1000. The highest accuracies are bolded, and the second-best is underlined. We observe that models tuned via prompts (MVP) are the most robust while preserving (or improving) the clean accuracy.

## 5 Results

We first evaluate the impact of using MVP on the adversarial robustness of NLP models. For the task of Boolean question answering (BoolQ), we find that fine-tuning a RoBERTa model with an MLP head (MLP-FT) achieves an accuracy of 28.2% on adversarial examples obtained through the TextFooler attack strategy (Table 1). Whereas, the corresponding accuracy for tuning the model via prompts (MVP) is 42.9% which is a considerable improvement over MLP-FT. Additionally, MVP leads to more robust models compared to adversarial training baselines like MLP-FT + Adv and InfoBERT that attain accuracies of 39.0% and 38.1% respectively. Further, MVP can be combined with adversarial training (MVP + adv), and doing so leads to an accuracy of 52.2% which is about a 10% improvement over MVP, without any loss in clean performance.

Similar to boolean question answering, the robustness advantages of MVP hold across the three tasks we examine. The individual performance statistics are detailed in Table 1. Overall, across four attack strategies, and three datasets, we report that MVP improves over MLP-FT by 8%. Remarkably, even in the absence of any adversarial training

MVP achieves the state-of-the-art adversarial performance improving baseline adversarial training methods by 3.5%. Moreover, it can be coupled with single-step adversarial training, resulting in an overall 7% improvement over state-of-art methods. Lastly, the robustness benefits come only at a 2x computation cost of standard training, as opposed to past works which need 5–10x computation cost of standard training due to additional adversarial training. Results on BERT-Base are in Table 7.

### 5.1 Sample Efficiency & Effective Robustness

We investigate the sample efficiency and effective robustness of MVP through experiments on the BoolQ and AG-News datasets using the RoBERTa-base model. We train models on randomly sampled fractions of the dataset, ranging from $5 \cdot 10^{-4}$ to $0.1$.

**Sample Efficiency** We compare the performance of MVP and MLP-FT in low-data regimes. We find that MVP results in models are consistently more robust compared to models trained through MLP-FT in the low data setups (see Figure 2a). In fact, we observe that in extremely low resource case (only 60 examples), it is hard to learn using MLP-FT, but model trained through MVP performs exceedingly well. We note that the relative bene-

|  |  |  | BoolQ | | | AGNews | | |
|---|---|---|---|---|---|---|---|---|
| **Experiment** | # Templates | Candidate | Clean | TFooler | TBugger | Clean | TFooler | TBugger |
| `MLP-FT` | N/A | N/A | $80.6 \pm 1.5$ | $28.2 \pm 1.7$ | $38.3 \pm 1.0$ | $94.5 \pm 0.4$ | $42.9 \pm 0.7$ | $61.8 \pm 0.3$ |
| Template Expansion | 1 | Class Label | $81.9 \pm 0.8$ | $35.9 \pm 0.2$ | $44.6 \pm 0.5$ | $94.6 \pm 0.4$ | $48.6 \pm 1.1$ | $67.3 \pm 1.1$ |
|  | 2 | Class Label | $82.3 \pm 0.2$ | $37.4 \pm 0.3$ | $46.4 \pm 0.5$ | $94.5 \pm 0.6$ | $50.8 \pm 1.6$ | $67.8 \pm 0.5$ |
|  | 3 | Class Label | $82.1 \pm 0.3$ | $40.8 \pm 1.5$ | $49.5 \pm 1.1$ | $94.2 \pm 0.2$ | $48.4 \pm 3.4$ | $66.2 \pm 1.1$ |
|  | 4 | Class Label | $82.0 \pm 0.6$ | $42.9 \pm 0.5$ | $49.8 \pm 1.6$ | $94.3 \pm 0.2$ | $51.4 \pm 2.0$ | $68.7 \pm 0.7$ |
| Candidate Exp. | 4 | Multiple | $81.6 \pm 1.2$ | $46.1 \pm 1.6$ | $53.0 \pm 0.7$ | $93.6 \pm 0.4$ | $54.0 \pm 0.7$ | $69.8 \pm 0.3$ |

Table 2: **Ablation Studies:** We study the impact of the number of candidate answers and prompt templates on adversarial performance of `MVP` (see §5.3). 'TFooler' and 'TBugger' represent model robustness under TextFooler and TextBugger attacks respectively. 'Clean' represents model accuracy on original test data. Additionally, we also assess the effect of including semantically similar answer candidates (see §6). All values are averaged over 3 seeds.

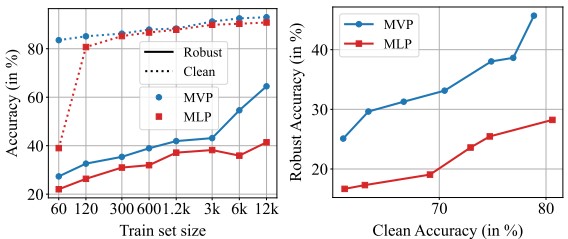

Figure 2: (a) **Sample Efficiency**: Clean and Robust Accuracy of RoBERTa-base model when trained using different data sizes of the AG News dataset. (b) **Effective Robustness**: Robust vs Clean Accuracy of RoBERTa-base model on the BoolQ dataset We find that (a) `MVP` is more sample efficient as compared to `MLP-FT`, and (b) `MVP` yields more robustness compared to `MLP-FT` for the same clean accuracy (see §5.1 for details).

fit of `MVP` over `MLP-FT` peaks around 5–10% of the data. Interestingly, the model trained through `MVP` requires only 5% of samples to achieve similar robustness levels as models trained with `MLP-FT` on the full dataset. In addition to robustness benefits, we find that `MVP` achieves considerably higher clean accuracy in low-data regimes (i.e., with $< 200$ examples). Results on BoolQ are in C.3.

**Effective Robustness** Effective robustness (Taori et al., 2021) measures the robust accuracy of models that have the same clean accuracy. This can help determine which training time design decisions will be valuable when scaled up. We measure the effective robustness for models trained through `MVP` and `MLP-FT` by training them on different data sizes. We find that even when both `MLP-FT` and `MVP` achieve the same clean accuracy, models trained through `MVP` are more robust (Figure 2b). Results on AG News are presented in C.3.

## 5.2 Out of Distribution Robustness

Going beyond adversarial robustness, we now perform experiments to assess the out-of-distribution robustness of `MVP`, `MLP-FT`, and `LPFT`. We use 5 sentiment classification datasets, namely SST2, Amazon Polarity (Zhang et al., 2016), IMDb (Maas et al., 2011), Movie Rationales (Zaidan et al., 2008), and Rotten Tomatoes (Pang and Lee, 2005). We fine-tune a Roberta model on 1000 examples of each of these datasets and evaluate all the datasets. Since all of these datasets are binary sentiment analysis datasets, we use the same template and candidate words across all the models (for both training and evaluation). Based on our investigation, we see that across 5 different models (and 20 evaluations) the average accuracy for `MVP` (89.65%) is 2% more than `MLP-FT` and 1.3% more than that of LPFT.

These results in Table 3 show that `MVP` is superior to `MLP-FT` and `LPFT` for both adversarial and OOD robustness. In summary, `LPFT` helps reduce the impact of random parameter vulnerability, but `MVP` additionally allows pre-training task alignment (the second hypothesis) hence resulting in superior performance and no fundamental trade-off be it OOD or adversarial robustness.

## 5.3 Ablation Studies

**Number of Candidate Answers** A larger candidate answer set is shown to improve clean performance in the few-shot setting (Hu et al., 2022). Here, we investigate the impact of the size of the candidate answer set on the adversarial performance of models tuned via prompts. The adversarial accuracy of the model with a single candidate answer is 42.9%, and it increases to 46.2% upon using an answer set com-

| Train v/s Eval | SST2 | | | Amazon Polarity | | | IMDb | | | Movie Rationales | | | Rotten Tomatoes | | |
|---|---|---|---|---|---|---|---|---|---|---|---|---|---|---|---|
| | MVP | MLP-FT | LPFT | MVP | MLP-FT | LPFT | MVP | MLP-FT | LPFT | MVP | MLP-FT | LPFT | MVP | MLP-FT | LPFT |
| SST2 | 91.3 | 91.2 | 91.9 | 92.8 | 89.2 | 90.1 | 89.4 | 87.6 | 87.9 | 86.0 | 85.9 | 86.2 | 86.1 | 83.2 | 84.1 |
| Amazon Polarity | 90.9 | 88.5 | 89.0 | 92.9 | 92.9 | 93.4 | 92.0 | 91.2 | 91.1 | 85.9 | 83.9 | 84.2 | 86.1 | 83.3 | 84.5 |
| IMDb | 84.4 | 81.4 | 83.5 | 91.9 | 88.8 | 88.7 | 92.2 | 91.9 | 92.4 | 92.0 | 89.9 | 90.2 | 81.9 | 78.1 | 80.1 |
| Movie Rationales | 89.9 | 85.9 | 85.4 | 92.5 | 89.1 | 90.7 | 91.7 | 90.6 | 91.6 | 94.4 | 93.5 | 94.3 | 87.4 | 83.0 | 83.4 |
| Rotten Tomatoes | 92.4 | 92.1 | 92.9 | 92.6 | 89.5 | 90.4 | 90.9 | 88.6 | 90.2 | 86.4 | 83.9 | 84.7 | 87.2 | 87.1 | 87.2 |
| Average | 89.8 | 87.8 | 88.5 | 92.5 | 89.9 | 90.7 | 91.3 | 90.0 | 90.6 | 89.0 | 87.4 | 87.9 | 85.7 | 83.0 | 83.9 |

Table 3: **OOD Robustness:** The results report the standard accuracy (in %) of a model trained on the dataset in the left-most column, and evaluated on 5 different OOD datasets. We see that across 5 different models (and 20 evaluations), the average accuracy for MVP (89.65%) on OOD tasks is 2% more than MLP-FT and 1.3% more than LPFT.

prising 4 candidates.[3] These results correspond to the RoBERTa-base model on BoolQ dataset against adversarial perturbations from the TextFooler attack. Overall, we observe an improvement of 1.0–3.5% in adversarial accuracy when we use a larger candidate set across different settings (Table 2). A more detailed analysis of the same with a single prompt template is provided in Appendix D.4.

**Number of Prompt Templates** Another design choice that we consider is the number of prompt templates used for prediction. We conjecture that the adversary may find it difficult to flip the model prediction when we average logits across multiple templates. To evaluate this, we train MVP with different number of prompt templates (ranging from 1 to 4), and compare the adversarial robustness. We notice a steady improvement in the adversarial accuracy as we increase the number of templates which supports our initial conjecture (see Table 2). While increasing the number of templates improves the robustness of the downstream model, MVP achieves large robustness gains even with a single template (compared to MLP-FT). Hence, using multiple prompt templates is not the fundamental reason for the improved robustness of MVP. Further, in order to assess the impact of the 'choice' of prompt templates used, we perform a more details analysis on the impact of prompt tuning for adversarial robustness of MVP in Appendix D.2. We find that even empty or random templates perform nearly similar to well-crafted prompts, and retain the robustness advantages of MVP over MLP-FT.

## 6   Why Does MVP Improve Robustness?

We test three hypotheses to explain the robustness gains achieved by MVP compared to MLP-FT in the context of adversarial attacks.

[3]Details about candidates and templates are in Appendix A

**Random Parameter Vulnerability** One plausible explanation for the observed adversarial vulnerability of MLP-FT is the randomly-initialized linear head used for downstream classification. The intuition behind this effect is that *fine-tuning a set of randomly-initialized parameters may lead to feature distortion of the pretrained model* as is demonstrated in Kumar et al. (2022). This phenomenon has also been observed in CLIP models (Radford et al., 2021), where the authors found that fine-tuning the model using a randomly initialized linear prediction head reduces the out-of-distribution robustness of the model. The phenomenon is unexplored in the context of adversarial robustness. We study this effect through three experiments.

1. ProjectCLS:   First, we reduce the number of random parameters by removing the dense layer of weights ($768 \times 768$ parameters) from the standard MLP architecture. We call this ProjectCLS, and only use a projection layer of dimensions $768 \times C$ parameters, with $C$ being the number of classes (see Figure 3(a)). We find that ProjectCLS is on average $\sim 8\%$ more robust than MLP-FT which suggests that reducing the number of randomly initialized parameters helps to increase model robustness (see Table 4).

2. CLSPrompt:   Second, we train another model, CLSPrompt, where instead of using the probabilities corresponding to the [MASK] token as in MVP, we use the probabilities of the candidate answers corresponding to the [CLS] token (see Figure 3(b)). The key difference between CLSPrompt and MLP-FT is that there are no randomly initialized MLP parameters in CLSPrompt, and we use the probabilities corresponding to the candidate answers, instead of projecting the representations with new parameters. From Table 4, we observe that CLSPrompt is once again on average $\sim 8\%$ more robust than MLP-FT which provides strong evidence in favor of our hypothesis of ran-

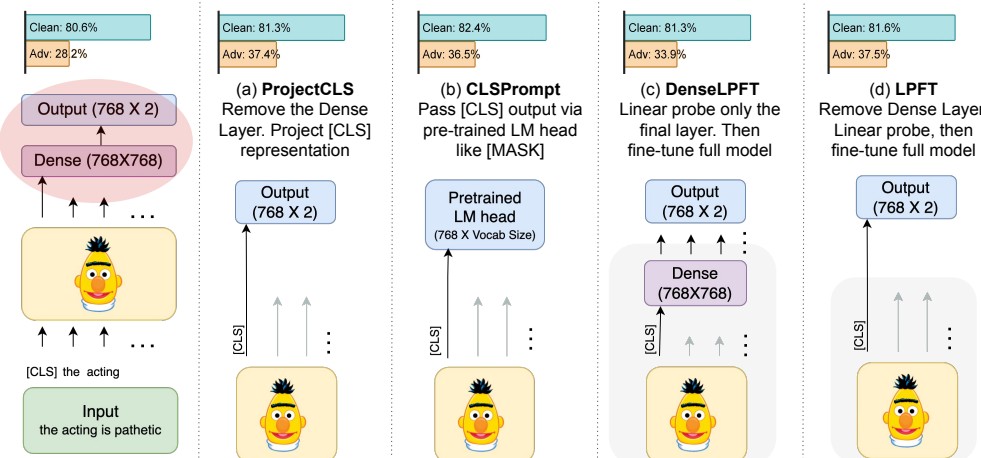

Figure 3: Various model tuning strategies for RoBERTa model trained on the BoolQ dataset. The corresponding clean and robust accuracies (under TextFooler attack) are also shown above each model paradigm. The left-most diagram shows the standard fine-tuning paradigm of `MLP-FT`, and each subsequent column modifies the architecture, helping us confirm the hypothesis that randomly initialized parameters are a cause of vulnerability.

| | | BoolQ | | | AGNews | | |
|---|---|---|---|---|---|---|---|
| **Hypothesis** | Setting | Clean | TFooler | TBugger | Clean | TFooler | TBugger |
| | `MLP-FT` | $80.6 \pm 1.5$ | $28.2 \pm 1.6$ | $38.3 \pm 1.0$ | $94.5 \pm 0.4$ | $42.8 \pm 0.7$ | $61.8 \pm 0.3$ |
| Random Parameter | `ProjectCLS` | $81.3 \pm 0.5$ | $37.4 \pm 1.2$ | $45.6 \pm 1.2$ | $93.7 \pm 0.4$ | $46.7 \pm 1.3$ | $65.2 \pm 3.3$ |
| | `CLSPrompt` | $82.4 \pm 0.3$ | $36.5 \pm 0.4$ | $46.0 \pm 1.2$ | $94.7 \pm 0.2$ | $47.2 \pm 1.9$ | $66.7 \pm 2.0$ |
| | `DenseLPFT` | $81.3 \pm 0.4$ | $33.9 \pm 1.4$ | $42.6 \pm 1.2$ | $94.5 \pm 0.6$ | $44.2 \pm 0.8$ | $64.5 \pm 1.1$ |
| | `LPFT` | $81.6 \pm 1.2$ | $37.5 \pm 1.1$ | $46.4 \pm 1.2$ | $94.5 \pm 0.1$ | $46.5 \pm 0.9$ | $67.2 \pm 1.0$ |
| Task Alignment | Untrained `MVP` | $67.5 \pm 0.9$ | $11.7 \pm 2.7$ | $14.9 \pm 2.7$ | $90.1 \pm 0.8$ | $12.2 \pm 2.9$ | $20.6 \pm 2.2$ |
| | Untrained `MLP-FT` | $67.0 \pm 0.6$ | $14.8 \pm 4.3$ | $17.5 \pm 1.1$ | $89.5 \pm 0.4$ | $13.4 \pm 1.2$ | $19.4 \pm 0.8$ |
| Candidate Semantics | Random (`MVP`) | $80.9 \pm 0.3$ | $42.1 \pm 0.4$ | $48.1 \pm 2.2$ | $93.4 \pm 0.3$ | $50.3 \pm 1.2$ | $68.3 \pm 0.3$ |

Table 4: Adversarial performance of RoBERTa for experiments corresponding to the random parameter vulnerability and task alignment hypotheses averaged over 3 seeds (§6). 'TFooler' and 'TBugger' represent model robustness under TextFooler and TextBugger attacks respectively. 'Clean' represents model accuracy on original test data.

dom parameter vulnerability.

3. LPFT (linear probe, then fine-tune): For our third experiment, we train two new models namely `LPFT` and `DenseLPFT` (see Figure 3(c,d)). In both these models, we do the following: (i) fit a logistic regression to the hidden states corresponding to the `[CLS]` token (linear probing); (ii) initialize the final layer of the classification head with the learned $768 \times C$ (where $C$ is the number of classes) matrix of the fitted logistic regression model; and (iii) fine-tune the whole model as in `MLP-FT`. The only difference between `LPFT` and `DenseLPFT` is that `DenseLPFT` has an additional randomly initialized dense layer of dimensions $768 \times 768$ unlike `LPFT`. In contrast to Kumar et al. (2022), we test `LPFT` against adversarial manipulations. We note from Table 4 that `DenseLPFT` is more robust than

`MLP-FT` (by over 10%) but it demonstrates lower robustness as compared to `LPFT` (by over 2%). This provides further evidence that randomly initialized parameters add to the vulnerability.

**Pretraining Task Alignment** The task of mask infilling aligns more naturally with the pretraining objective of the language model and we posit that finetuning via mask infilling as in `MVP` results in robustness gains. To test this hypothesis, we use an untrained RoBERTa model, and measure the clean accuracy and robustness of `MVP` and `MLP-FT` models. We observe that in the absence of pre-training, `MVP` trained with a single template does not achieve any additional robustness over the baseline, and in fact, `MLP-FT` performs better than `MVP` (Table 4) whereas in the presence of pre-training, `MVP` outperforms `MLP-FT` (Table 2) in

all the settings. Note that this does not contradict the hypothesis about vulnerability due to randomly-initialized parameters, as that hypothesis only applies for pretrained models. This suggests that the alignment of `MVP` with the pre-training task is crucial for adversarial robustness on downstream task.

**Semantically Similar Candidates** We question whether the improvement in robustness can also be attributed to the semantic relatedness between candidate answers and the class labels. To answer this question, we change the candidate answers to random proper nouns ('jack', 'john', 'ann', 'ruby') for the 4-class classification problem of AG-News and ('jack', 'john') for the 2-class classification task of BoolQ. All of these words are unrelated to the class labels. We find that irrespective of whether we use semantically related candidates or not, the robust accuracy of the model is within 1% of each other, thereby implying that using semantically similar candidates is not a factor behind the robustness gains of `MVP` (Table 4). While the choice of candidate answers is crucial in the pre-train, prompt, and predict paradigm (Hu et al., 2022), it is irrelevant in the pre-train, prompt, and finetune paradigm. With sufficient fine-tuning over the downstream corpus, a model can learn to associate any candidate word with any class, irrespective of its semanticity.

However, one may wonder why using 'random' candidate words doesn't hurt the model robustness, since this also leads to modifying a 'parameter' in the model's embedding space, which was initially uncorrelated to the class label. We analyze this question in detail in Appendix D.3 and conclude that the main reason for the preserved robustness is the 'pre-training task hypothesis' and the fact that the modified word embeddings have a much smaller dimension of size 768 x C (where C is the number of candidate words), as opposed to modifying a dense layer.

## 7 Human Study

We conduct a human study to assess the viability of the adversarial attacks. More specifically, we provide machine learning graduate students 250 input examples and ask the following questions: (a) What is the perceived label of the sentence; (b) What is their confidence about this label; and (c) Was this sentence adversarially manipulated? We use the BoolQ dataset and strictly instruct our annotators to not use any external knowledge but the context of the given passage only. We use samples that were successfully attacked by TextFooler for `MVP + Adv`

model with a RoBERTa backbone. As a control for the study, we provide the original sentence rather than the adversarially perturbed one 33% times. The underlying model achieves a clean accuracy of 81.7% and a robust accuracy of 54.0%.

We find that human annotators identify 29% of adversarial examples to be perturbed as opposed to only 6% of clean examples. Moreover, we also discover that humans achieved 11% lower accuracy on adversarial examples as compared to clean examples (85% → 74%) with average confidence on the label of perturbed examples being 15% lower (90% → 75%). This study highlights that a fraction of adversarial attacks either manipulate the input so significantly that it is easily detectable, or change the label, signifying that `MVP` is more robust than what crude statistics suggest in §5. Details related to the human study are available in Appendix F.1.

## 8 Conclusion

In this work, we benchmark the robustness of language models when adapted to downstream classification tasks through prompting. Remarkably, model tuning via prompts—which does not utilize any sort of adversarial training or prompt engineering—already outperforms the state-of-the-art methods in adversarially robust text classification by over 3.5% on average. Moreover, we find that `MVP` is sample efficient and also exhibits high *effective* robustness as compared to the conventional approach of fine-tuning with an MLP head (`MLP-FT`). We find that the lack of robustness in baseline methods can largely be attributed to the lack of alignment between pre-training and finetuning task, and the introduction of new randomly-initialized parameters.

## 9 Limitations

This work considers models that are under 1B parameters in size. While larger models are becoming popular in the NLP community, developing practical attacks that scale to such large models is an extremely challenging task. For instance, for the evaluation considered in this paper, each attack takes approximately a day on a single A6000 GPU to run (across multiple seeds of the model). Furthermore, the scope of our work is limited to tasks where fine-tuning with an MLP head is commonplace. This includes boolean question answering, sentence classification, and paraphrase detection tasks. Finally, using multiple templates for `MVP` comes with a trade-off with latency which is discussed in Appendix D.1.

**Broader Impact** Our work does not pose any immediate negative impacts to society, except for the carbon emissions owing to the training and evaluation of big models. We emphasize that the adversarial robustness conferred via `MVP` is a desirable property for deployed systems, and our work contributes towards making NLP models more reliable and safe when deployed in real-world settings.

## Acknowledgements

We thank the Pittsburgh weather that kept us from doing anything but work on this draft. PM is supported by funding from the DARPA GARD program. ZL acknowledges the NSF (FAI 2040929 and IIS2211955) Amazon AI, UPMC, Highmark Health, Abridge, Ford, Mozilla, the PwC Center, the Block Center, the Center for Machine Learning and Health, and the CMU Software Engineering Institute (SEI) via Department of Defense contract FA8702-15-D-0002, for their generous support of ACMI Lab's research. DP is grateful for the support of Adobe Inc., Google and Kotak IISc AI-ML Centre (KIAC).

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

# Supplementary Material
# Model-tuning Via
# Prompts Makes NLP Models More Robust

## A  Candidate Answers & Prompt Templates

We enumerate all the prompt templates and candidate answers used for our experiments on MVP. Templates beginning with the [SEP] token are appended at the end of the input otherwise they precede the input. Note that we remove the [SEP] token and then append the template to the input. The [SEP] token is just used as an indicator for appending the template to the input. Note that since Causal Language models are not bidirectional, for GPT-2 experiments, all the prompt templates will be appended at the end of the input.

**AG News**   The prompt templates used for MLMs:

1. A [MASK] news

2. [SEP] This topic is about [MASK]

3. Category : [MASK]

4. [SEP] The category of this news is [MASK]

The prompt templates used for GPT-2 are:

1. [SEP] This topic is about [MASK]

2. [SEP] The category of this text is [MASK]

3. [SEP] Category : [MASK]

4. [SEP] This is a news from [MASK]

The candidate answers used are the same as the class labels, namely—politics, business, sports, and technology—for all the experiments except the larger candidate set ablation study. For that ablation, we use the following candidate answer set:

1. {politics, world, government, governance}

2. {sports, competition, games, tournament}

3. {business, corporation, enterprise, commerce}

4. {technology, science, electronics, computer}

**BoolQ**   The prompt templates used for MLMs are:

1. Answer to the question is [MASK]

2. [SEP] [MASK]

3. I think [MASK]

4. [SEP] The answer is [MASK]

The prompt templates used for GPT-2 are the same as above except every template is appended to the end of the input. As in AG News, the candidate answers used are the same as the class labels, namely false and true, except when performing the larger candidate set experiment, in which case we use the following candidate answer set:

1. {false, wrong, incorrect, invalid}

2. {true, correct, valid, accurate}

**SST-2**   The prompt templates used for MLMs are:

1. Sentiment of the statement is [MASK] .

2. [SEP] [MASK]

3. This is a [MASK] statement

4. [SEP] The statement is [MASK]

Similar to AG News and BoolQ, we use the class labels (i.e., negative and positive) as the candidate answers.

**DBPedia14**   The prompt templates used for MLMs are:

1. Content on [MASK]

2. [SEP] This topic is about [MASK]

3. Category : [MASK]

4. [SEP] The content is about [MASK]

The candidate answers used are:
{0: 'company', 1: 'education', 2: 'artist', 3: 'athlete', 4: 'office', 5: 'transportation', 6: 'building', 7: 'nature', 8: 'village', 9: 'animal', 10: 'plant', 11: 'album', 12: 'film', 13: 'writing'}

**MRPC** The prompt templates used for MLMs:

1. The two sentences are `[MASK]`

2. `[SEP]` First sentence is `[MASK]` to second sentence

3. Two `[MASK]` sentences

4. `[SEP]` The two sentences have `[MASK]` meanings

The candidate answers used are:
{0: 'different', 1: 'equivalent'}

## B  Baseline Methods and Attacks

### B.1  Baselines

We describe training schemes corresponding to various fine-tuning strategies below.

**MLP-FT** : This is the "base" model for classification via standard non-adversarial training and is utilized by all the baselines discussed in this section. Given a pre-trained model, we perform downstream fine-tuning by adding an MLP layer to the output corresponding to `[CLS]` token as illustrated in Figure 1(a). This hidden representation is of size $768 \times 1$. In the case of the BERT model, there is a single dense layer of dimension $768 \times 2$, whereas in the case of RoBERTa model, we have a two-layer MLP that is used to make the final prediction.

**MLP-FT + Adv**: This is identical to the method used for adversarial training in Section 3.2, wherein we perform adversarial perturbations in the embedding space of the `MLP-FT` model, rather than `MVP`.

**FreeLB++** (Li et al., 2021): Free Large-Batch (FreeLB) adversarial training (Zhu et al., 2020) performs multiple Projected Gradient Descent (PGD) steps to create adversarial examples, and simultaneously accumulates parameter gradients which are then used to update the model parameters (all at once). FreeLB++ improves upon FreeLB by increasing the number of adversarial training steps to 10 and the max adversarial norm to 1.

**InfoBERT** (Wang et al., 2021a): InfoBERT uses an Information Bottleneck regularizer to suppress noisy information that may occur in adversarial attacks. Alongside, an 'anchored feature regularizer' tries to align local stable features to the global sentence vector. Together, this leads to improved generalization and robustness. InfoBERT can additionally be combined with adversarial training (we use Free LB++ for this purpose).

**AMDA** (Si et al., 2021b): Adversarial and Mixup Data Augmentation (AMDA) improves robustness to adversarial attacks by increasing the number of adversarial samples seen during training. This method interpolates training examples in their embedding space to create new training examples. The label assigned to the new example is the linear interpolation of the one hot encodings of the original labels.

### B.2  Attack Details

In the main paper, we evaluated our method on three popular word substitution attacks and one character-level attack. These included the TextFooler, TextBugger and BertAttack attack strategies. TextFooler and TextBugger are word substitution attacks that replace words with "similar" neighboring words (where similarity is based on counterfitted GloVe embeddings). TextFooler greedily searches in a large set of neighbors (in the embedding space) for each word, so long as they satisfy some constraints on embedding similarity and sentence quality. An additional constraint requires the substituted word to match the POS of the original word. TextBugger, on the other hand, restricts the search space to a small subset of neighboring words and only uses sentence quality as a constraint. To control the amount of change made by an attack, we limit the adversary to perturbing a maximum of 30% words in the AG News dataset and 10% in all other datasets. We do not modify any other constraints (such as the query budget) and run the attacks on 1000 examples from the test set. We also evaluate on one character-level, and another word substitution attack. For character-level attack, we use the adversarial misspellings attack introduced by Pruthi et al. (2019), and we additionally evaluate the popular BertAttack (Li et al., 2020).

## C  Extended Experiments on Adversarial Robustness

### C.1  Results on Additional Datasets and Models

**Results on BERT-Base** Results on BERT-Base model are presented in Table 7. Similar to the results corresponding to RoBERTa-Base model in the main paper, we find that our proposed method `MVP` improves over the state-of-art defenses across 3 different datasets and 4 different attacks by 2% even without any adversarial training. Using adversarial training further improves the average robust accuracy by 4%.

| | GPT2 | | | | | |
|---|---|---|---|---|---|---|
| | BoolQ | | | AG News | | |
| | Clean Acc | TextFooler | TextBugger | Clean Acc | TextFooler | TextBugger |
| MLP-FT | $61.0 \pm 2.1$ | $20.2 \pm 0.6$ | $24.9 \pm 1.4$ | $93.7 \pm 0.2$ | $27.6 \pm 1.2$ | $58.2 \pm 0.9$ |
| MLP-FT +Adv | $60.5 \pm 0.4$ | $22.0 \pm 1.1$ | $31.8 \pm 1.8$ | $92.4 \pm 0.3$ | $\underline{39.6 \pm 0.5}$ | $61.3 \pm 0.7$ |
| MVP | $72.5 \pm 1.0$ | $\underline{28.7 \pm 1.6}$ | $\underline{38.3 \pm 1.6}$ | $93.8 \pm 0.3$ | $31.4 \pm 0.5$ | $\underline{61.0 \pm 0.8}$ |
| MVP +Adv | $71.8 \pm 0.8$ | $\mathbf{30.1 \pm 0.6}$ | $\mathbf{41.2 \pm 0.8}$ | $93.7 \pm 0.3$ | $\mathbf{44.0 \pm 0.2}$ | $\mathbf{64.4 \pm 1.2}$ |

Table 5: Adversarial Robustness results on BoolQ and AG News dataset using GPT-2 model. All experiments are run on 3 different seeds and the performance is reported over a fixed test set of size 1000. The best-performing robust accuracies are bolded and the second best robust accuracies are underlined.

| | DBPedia | | | | |
|---|---|---|---|---|---|
| | Clean Acc | TextFooler | TextBugger | BertAttack | Misspellings |
| MLP-FT | $97.3 \pm 0.7$ | $43.8 \pm 1.5$ | $68.7 \pm 0.9$ | $72.4 \pm 1.2$ | $65.7 \pm 1.3$ |
| MLP-FT + Adv | $97.2 \pm 0.4$ | $56.1 \pm 0.2$ | $76.4 \pm 0.3$ | $78.3 \pm 0.6$ | $72.2 \pm 0.7$ |
| MVP | $97.0 \pm 0.5$ | $\underline{57.2 \pm 1.0}$ | $\underline{77.2 \pm 0.5}$ | $\underline{80.6 \pm 0.7}$ | $\underline{74.3 \pm 0.7}$ |
| MVP + Adv | $97.3 \pm 0.9$ | $\mathbf{82.7 \pm 0.4}$ | $\mathbf{90.3 \pm 0.2}$ | $\mathbf{88.5 \pm 1.8}$ | $\mathbf{86.4 \pm 0.3}$ |
| | MRPC | | | | |
| | Clean Acc | TextFooler | TextBugger | BertAttack | Misspellings |
| MLP-FT | $87.9 \pm 0.6$ | $41.5 \pm 1.2$ | $50.2 \pm 1.0$ | $61.1 \pm 1.1$ | $51.7 \pm 1.0$ |
| MLP-FT + Adv | $87.2 \pm 0.4$ | $42.1 \pm 0.3$ | $53.4 \pm 0.7$ | $64.1 \pm 0.1$ | $54.2 \pm 0.4$ |
| MVP | $88.4 \pm 0.4$ | $\underline{44.8 \pm 0.1}$ | $\underline{56.6 \pm 0.1}$ | $\underline{68.8 \pm 0.5}$ | $\underline{57.3 \pm 0.9}$ |
| MVP + Adv | $87.1 \pm 1.2$ | $\mathbf{46.6 \pm 1.2}$ | $\mathbf{60.7 \pm 0.4}$ | $\mathbf{72.1 \pm 0.9}$ | $\mathbf{65.8 \pm 0.3}$ |

Table 6: Adversarial performance of RoBERTa-base model on 2 additional datasets. All accuracy values are reported for a fixed test set of size 1000 and are averaged over 3 different seeds. The highest accuracies are bolded, and the second-best is underlined. MVP is the most robust, and preserves (or improves) the clean accuracy.

**Additional Datasets** We further extend our results on two diverse datasets—DBPedia14 (Zhang et al., 2015a), a 14-class news classification dataset, and MRPC (Dolan and Brockett, 2005), a paraphrase detection dataset. Results on these are presented for the MLP-FT and MVP training schemes for RoBERTa-base model in Table 6.

The experiments provide additional evidence to support our findings about the adversarial robustness conferred by model-tuning via prompts (MVP) as opposed to the conventional approach of MLP-FT. Without adversarial training, MVP improves over MLP-FT by an average of 6% on the MRPC dataset across 4 different attacks. Results on the DBPedia dataset also show consistent improvements of MVP over MLP-FT . In particular, we find that MVP improves on average (across 4 different attacks) by 10% over MLP-FT, and MVP + adv improves by 16% over the adversarial training counterpart of MLP-FT. In a setting where the number of labels is many, we in fact see a larger relative gain by using MVP over the conventional approach of MLP-FT.

## C.2 Results on Causal Language Models

Causal Language Models have not been traditionally fine-tuned for downstream classification tasks. This is evident also from the exclusion of fine-tuning results in the original GPT-2 paper (Radford et al., 2019). In this work, we try to evaluate the clean and adversarial robustness of GPT-2 models, when adapted to downstream tasks. To implement MVP, we use the Causal Language Modeling (CLM) head to get the next word prediction logits. Since we are using the CLM head, it is imperative that the prompt templates are appended at the back and have the [MASK] as the last token.

We find that on the BoolQ dataset MLP-FT achieves a robust accuracy of 20.2% and MVP achieves a robust accuracy of 28.7% (Table 5), which is a large improvement. Similar to our findings in the main paper, 1-step adversarial training on MVP (MVP + Adv) yields a robust accuracy of 30.1% which is a massive improvement over MLP-FT and MLP-FT + Adv which obtains a robust accuracy of 22.0%. Interestingly, we also

| | MLP-FT | MLP-FT + Adv | Free LB++ | MADA | InfoBert | MVP | MVP + Adv |
|---|---|---|---|---|---|---|---|
| | | | | SST2 | | | |
| Clean Acc | 91.9 ±0.2 | 90.9 ±0.3 | 92.1 ±0.8 | 92.1 ±0.9 | 91.7 ±0.6 | 91.7 ±0.4 | 91.8 ±0.7 |
| TextFooler | 38.3 ±1.0 | 42.8 ±1.2 | 42.2 ±1.0 | 41.7 ±0.5 | 43.1 ±0.8 | 44.6 ±0.7 | **47.7 ±0.6** |
| TextBugger | 60.4 ±0.4 | 62.3 ±0.5 | 63.0 ±0.7 | 60.9 ±0.4 | 64.6 ±0.6 | 65.1 ±0.1 | **67.8 ±0.4** |
| Bertattack | 68.7 ±0.5 | 70.1 ±0.8 | 72.0 ±0.9 | 70.3 ±0.7 | 72.8 ±0.6 | 75.9 ±0.7 | **78.9 ±0.9** |
| Misspellings | 39.2 ±0.4 | 42.4 ±0.4 | 43.4 ±0.4 | 40.2 ±0.7 | 43.1 ±0.7 | 45.6 ±1.1 | **49.2 ±0.9** |
| | | | | AG News | | | |
| Clean Acc | 93.7 ±0.4 | 93.2 ±0.2 | 93.4 ±0.2 | 92.8 ±0.5 | 93.8 ±0.3 | 93.7 ±0.5 | 94.0 ±0.6 |
| TextFooler | 37.5 ±0.7 | 44.3 ±1.0 | 43.5 ±0.2 | 41.8 ±0.9 | 44.0 ±1.6 | 46.3 ±1.2 | **53.7 ±0.1** |
| TextBugger | 58.9 ±0.6 | 64.1 ±0.2 | 63.4 ±0.8 | 62.6 ±1.0 | 64.1 ±0.8 | 66.0 ±0.4 | **69.2 ±1.3** |
| Bertattack | 78.1 ±1.2 | 80.1 ±0.2 | 80.9 ±0.1 | 79.6 ±0.6 | 80.7 ±0.6 | 82.1 ±0.7 | **83.4 ±0.4** |
| Misspellings | 76.8 ±0.8 | 78.5 ±0.2 | 79.5 ±0.7 | 76.9 ±1.3 | 79.6 ±0.7 | 81.5 ±0.4 | **84.3 ±0.3** |
| | | | | BoolQ | | | |
| Clean Acc | 71.1 ±1.3 | 71.0 ±0.9 | 70.7 ±0.2 | 71.1 ±0.9 | 71.8 ±0.6 | 71.4 ±1.0 | 71.3 ±0.3 |
| TextFooler | 21.8 ±4.4 | 29.8 ±0.8 | 29.5 ±0.6 | 25.4 ±0.8 | 29.9 ±0.2 | 31.1 ±1.3 | **43.1 ±0.7** |
| TextBugger | 36.8 ±3.0 | 42.8 ±1.3 | 42.8 ±0.6 | 41.6 ±0.6 | 42.6 ±0.6 | 44.4 ±2.8 | **49.9 ±0.9** |
| Bertattack | 55.7 ±1.2 | 57.8 ±0.7 | 58.2 ±0.9 | 57.8 ±0.6 | 58.9 ±0.8 | 60.1 ±0.6 | **63.2 ±0.7** |
| Misspellings | 55.1 ±1.0 | 58.1 ±0.3 | 59.4 ±0.7 | 56.2 ±0.7 | 59.1 ±0.6 | 60.1 ±1.0 | **63.2 ±0.8** |

Table 7: Adversarial performance of BERT-base model on 3 different datasets. All accuracy values are reported for a fixed test set of size 1000 and are averaged over 3 different seeds. The highest accuracies are bolded, and the second-best are underlined. MVP is the most robust, and preserves (or improves) the clean accuracy.

notice that for MLP-FT and MLP-FT + Adv, it is difficult to achieve a good clean generalization performance whereas MVP and MVP + Adv perform much better on the clean test set. These observations are in line with the results in our main paper. On the AG News dataset, MVP performs significantly better than MLP-FT and MVP + Adv performs better than MLP-FT + Adv. These results show that MVP is not only a good way of finetuning BERT-like MLMs but can also improve Causal Language Models both in terms of clean accuracy and robustness to adversarial perturbations.

## C.3 Sample Efficiency and Effective Robustness

We demonstrate the sample efficiency of MVP on the BoolQ dataset (Figure 4a) in addition to the discussion about AG News in §5.1. Interestingly we find that MLP-FT is unable to achieve better accuracy compared to even random classifiers with 200 examples but MVP performs much better in the low data regime (< 200 examples). We also provide more evidence on the effective robustness of MVP by presenting the effective robustness results on AG News (Figure 4b). Even for AG News, we notice that the curve is much steeper for MVP than MLP-FT.

| Configuration | Time taken (in sec) |
|---|---|
| MLP-FT | $23.25 \pm 0.37\ (0.95 \times T)$ |
| 1 Template | $24.45 \pm 0.25\ (T)$ |
| 2 Templates | $27.51 \pm 0.45\ (1.15 \times T)$ |
| 3 Templates | $32.81 \pm 0.30\ (1.34 \times T)$ |
| 4 Templates | $35.65 \pm 0.44\ (1.45 \times T)$ |

Table 8: Inference latency comparison across different configurations.

## D Extended Analysis

### D.1 Latency of Using Multiple Templates

We present the latency numbers and compare them with the latency of the standard MLP-FT approach. Specifically, the time required for 2000 forward passes of data from the IMDb dataset with a batch size of 1 is represented as $T = 24.45 \pm 0.25$ seconds. The results are presented in Table 8.

In summary, using multiple templates makes predictions about 1.45x slower, however, this leads to improved robustness.

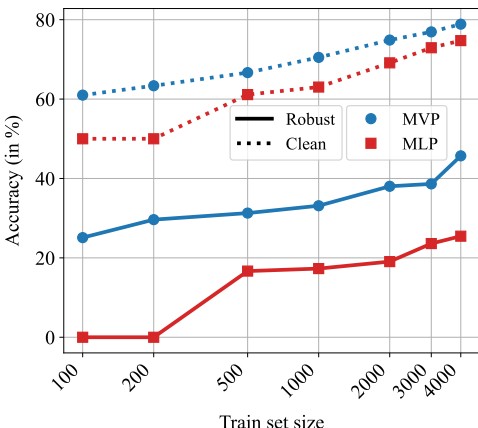

(a) Clean and adversarial accuracies of RoBERTa-base model on BoolQ dataset for varying amounts of training data.

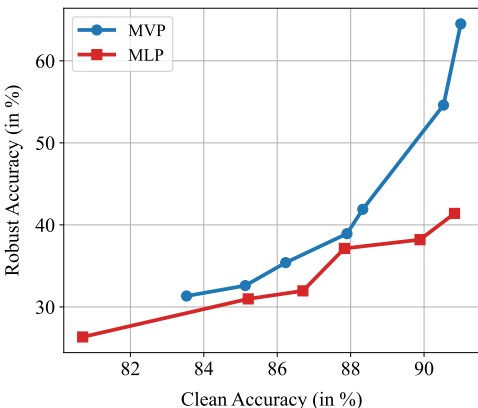

(b) Clean vs adversarial performance of RoBERTa base model for the AG News dataset. We find that models tuned via prompts (MVP) yield more robust models compared to fine-tuning MLP heads for the same clean accuracy.

Figure 4: (a) Models trained with MVP are significantly more sample efficient as compared to those with MLP-FT . (b) We find that models tuned via prompts (MVP) yield more robust models compared to fine-tuning MLP heads for the same clean accuracy (details in §5.1).

### D.2 Benefits from Prompt Tuning

To assess the benefit of Prompt tuning, we conducted a series of experiments. Interestingly, even an empty template with just a [MASK] token, which would be considered a weak prompt, showed significant performance improvements over the standard technique of MLP-FT. We present these results for 4 different prompt choices in Table 9. The choice of prompts used has very little effect on model robustness in the fine-tuning regime. We tabulate the robustness results corresponding to different prompts below (for the BoolQ dataset). Here the first four prompts are the prompts we used and "Ruby emerald [MASK]" is a random prompt from

vocabulary words.

We did not perform any dedicated prompt tuning for selecting the prompts. Instead, prompts were either chosen directly or inspired by the *OpenPrompt* repository. The selected prompts led to a marginal (2%) increase in model robustness during fine-tuning. Unlike typical few-shot or in-context learning methods, our approach aligns more closely with the idea of prompt tuning. For more advanced techniques and further potential improvements in prompt tuning, readers are referred to Hu et al. (2022).

### D.3 Why does using "dummy candidate words" not hurt model robustness?

In our paper, we note that using dummy candidate words like Jack and Ann, instead of class labels, does not hurt model robustness. However, this is very similar to random projection layers, so why does this not impact model robustness similarly? We note that using dummy candidate words leads to modifying an embedding of size 768 x $C$ (where $C$ is the number of candidate words) so that they now have a new "meaning". The effective number of "new parameters" is much lower than the parameters in the "dense 768x768 layer" in the Roberta model. However, in terms of new parameter complexity, this is similar to our ablation "ProjectCLS". As one may note, using ProjectCLS also improves robustness over MLP-FT. This is because we avoid the dense 768x768 layer.

Additionally, we conducted a new experiment of using empty slots in the vocabulary of Roberta and compared it with using "dummy candidate words" and "class labels". For the BoolQ dataset, using a Roberta model, we summarize the results in Table 10.

Based on these accuracies above we find that:

1. Using class labels is better than using "dummy/untrained words" for both clean and robust accuracy, which supports the random parameter vulnerability hypothesis.

2. The robustness achieved upon using completely untrained slots is similar to that when using dummy candidate words. This suggests that when compared to class labels, modifying dummy words has a similar loss in robustness as with modifying untrained words.

3. The MVP models (with random/empty candidate words) are more robust than

| Template | Clean Accuracy | Robust Accuracy |
|---|---|---|
| Answer to the question is `[MASK]` | 81.5±0.5 | 38.5±0.7 |
| `[SEP]` `[MASK]` | 81.7±0.6 | 36.1±0.4 |
| I think `[MASK]` | 81.9±0.8 | 35.9±0.2 |
| `[SEP]` The answer is `[MASK]` | 82.0±0.3 | 38.1±0.1 |
| `[SEP]` Ruby emerald `[MASK]` | 81.4 ± 0.5 | 36.8 ± 0.4 |
| None (`MLP-FT`) | 80.6±1.5 | 28.2±1.7 |

Table 9: Model robustness per template chosen for the BoolQ dataset.

| Method | Candidate Choice | Clean Accuracy | Robust Accuracy |
|---|---|---|---|
| `MLP-FT` | N/A | 80.6 ± 1.5 | 28.2 ± 1.6 |
| ProjectCLS | N/A | 81.3 ± 0.5 | 37.4 ± 1.2 |
| `MVP` | Class Labels | 82.0 ± 0.6 | 42.9 ± 0.5 |
| `MVP` | Dummy Candidate Words (Jack/Ann) | 80.9 ± 0.3 | 42.1 ± 0.4 |
| `MVP` | Empty Slots | 81.3 ± 0.3 | 41.2 ± 0.7 |

Table 10: Comparison of different choices of candidate words and their accuracies when training a Roberta model on the BoolQ dataset.

`ProjectCLS` (which already bridges the robustness gap from `MLP-FT`). These gains are explained by the pre-training task alignment hypothesis, where pre-training (and fine-tuning) the model with the task of `[MASK]` infilling helps make the downstream model robust.

### D.4  Impact of Ensembling the Candidates

Recall that in the main paper, we ensemble multiple templates and aggregate their predictions. In this subsection, we also investigate the impact of ensembling candidate words rather than templates. Based on the results in Table 11, we find that this is not as helpful as ensembling multiple templates.

### E  Hyperparameter Details

**Attack Hyperparameters** TextFooler and TextBugger use a word substitution attack that searches for viable substitutions of a word from a set of synonyms. We restrict the size of the synonym set to 50 for TextFooler which is the default value used by Jin et al. (2020) and to 5 which is the default value used by Li et al. (2018). Both TextFooler and TextBugger use a Universal Sentence Encoder (USE), that poses a semantic similarity constraint on the perturbed sentence. We use the default value of 0.84 as the minimum semantic similarity. Another important attack parameter is the maximum percentage of modified words ($\rho_{max}$). As discussed

in (Li et al., 2021), we use $\rho_{max} = 0.3$ for AG News and use $\rho_{max} = 0.1$ for BoolQ and SST2 in all our experiments. We use a query budget of 100 for BERT-Attack and a query budget of 300 for adversarial misspellings as these attacks are very slow.

**Training Hyperparameters & Model Selection** We train all models including the baselines with patience of 10 epochs, for a maximum of 20 epochs, and choose the best model based on validation accuracy. For the datasets that do not contain a publicly available validation set, we set aside 10% of the training set for validation. In the case of baseline defenses that use adversarial training, we perform model selection based on adversarial accuracy rather than clean accuracy. We use a candidate answer set containing only the class label names and we average over 4 prompt templates in all the `MVP` models. We use a batch size of 32 for `MLP-FT` and a batch size of 8 for `MVP` models. The learning rate is set as $1e-5$ for all the models. We use the AdamW optimizer along with the default linear scheduler (Wolf et al., 2020). In all the `MVP` + Adv and `MLP-FT` + Adv models, we use a use 1-step adversarial training with max $\ell_2$ norm of 1.0. For the state-of-the-art baselines, we use the same hyperparameters as prescribed by the original papers.

| Configuration | Clean Accuracy | Robust Accuracy |
|---|---|---|
| 1 prompt + 4 candidates | 81.9±0.3 | 37.4±0.5 |
| 1 prompt + 1 candidate | 81.9±0.8 | 35.9±0.2 |
| 4 prompt + 1 candidate | 82.0±0.6 | 42.9±0.5 |

Table 11: Impact of different ensembling configurations on clean and robust accuracy of Roberta model on the BoolQ dataset.

| | | Perturbed Examples | Unperturbed Examples |
|---|---|---|---|
| Q1. Annotator Accuracy | | 74% | 85% |
| Q2. Annotator Confidence | | 75% | 90% |
| | No | 54% | 82% |
| Q3. Perturbed? | Unsure | 17% | 12% |
| | Yes | 29% | 06% |

Table 12: Summary of the responses from the user study. The total number of presented examples is 250, out of which 83 are unperturbed and 167 are adversarially perturbed.

## F Human Study

Despite the improvements brought to adversarial robustness by our proposed modification (MVP + Adv), we note that there is still a significant drop in robust accuracy as opposed to the clean accuracy of the model. We conduct a human study in order to (i) assess the viability of adversarial attacks, and (ii) estimate human performance against adversarial attacks. More specifically, we provide machine learning graduate students 250 input examples and ask the following questions:

1. What is the perceived label of the sentence? (Answer options: True or False)

2. On a scale of 1 to 3, what is their confidence about this label?

3. Was this sentence adversarially manipulated? (Answer options: Yes, Unsure, or No)

We use the BoolQ dataset and strictly instruct our annotators to not use any external knowledge but the only context of the given passage. We use samples that were successfully attacked by TextFooler for MVP + Adv model with a RoBERTa backbone. As a control for the study, 33% of all sentences are unperturbed sentences from the original dataset. The underlying model achieves a clean accuracy of 81.7% and a robust accuracy of 54.0%.

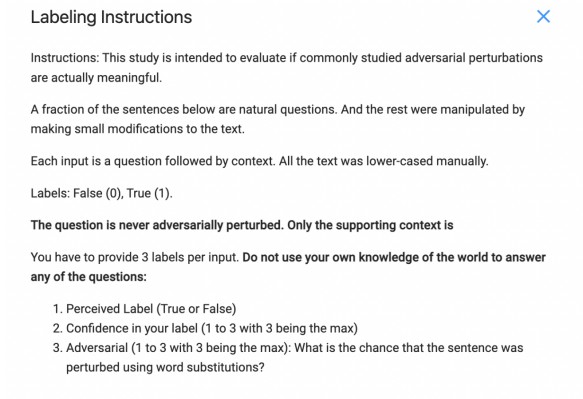

Figure 5: A snapshot of the instructions for completing our study.

First, we find that humans achieved 11% lower accuracy on adversarial examples as compared to clean examples (85% → 74%) with average confidence on the label of perturbed examples being 15% lower (90% → 75%) (Table 12). Next, we also discover that human annotators suspect 29% of adversarial examples to be perturbed as opposed to only 6% of clean examples. Through this study, we also find that in 47% of the cases, the input is either manipulated so significantly that it is easily detectable or the original label is not preserved, signifying that MVP may be more robust than what statistics suggest in §5. Further details are available in Appendix F.1.

### F.1 Details of Interface

We present a snapshot of our interface that provides detailed instructions for our users (Figure 5). We provide a detailed overview of the questions asked in the user study. Annotators were provided with a boolean question and an accompanying context to answer the question and asked were asked to annotate the following:

1. What should be the answer to the question? (only use the context) Given the boolean question and the context, we ask the annotators whether the answer to the question is True or False. We also

request the annotators only use the given context and refrain from using any external knowledge.

**2. How confident are you about the label above?**
Once the annotator has answered question 1, we ask them to rate how confident they feel about the label they assigned to the input. The options provided are "Uncertain", "Somewhat Certain" and "Certain". Based on their response we assign a confidence of 1, if the annotator was certain, assign 0.5 if the annotator was somewhat certain, and assign 0 if the annotator was uncertain to calculate the average confidence.

**3. Do you think that the sentence is adversarially perturbed? (using word substitutions) Do not use your own knowledge of the world to answer this question.** We also ask the annotators, if the input was adversarially perturbed. The options provided to the user are "No", "Unsure" and "Yes".

The annotators helped us annotate 250 such examples out of which 167 were adversarially perturbed and 83 were clean. An overview of the responses from this study is presented in Table 12, and the key takeaways are discussed in Section F.