# OpenReview forum: "Model-tuning Via Prompts Makes NLP Models Adversarially Robust"
_EMNLP/2023/Conference — EMNLP 2023 Main_

### Official Review · Reviewer_QuR4 · 2023-08-02

**Soundness:** 4

**Excitement:**

3: Ambivalent: It has merits (e.g., it reports state-of-the-art results, the idea is nice), but there are key weaknesses (e.g., it describes incremental work), and it can significantly benefit from another round of revision. However, I won't object to accepting it if my co-reviewers champion it.

**Paper Topic And Main Contributions:**

The paper proposes a novel prompt-based tuning strategy that helps increase the robustness of the model. Experimental results show that it can outperform previous sota methods.

**Reasons To Accept:**

- The idea of prompt tuning for robustness is interesting. The author states the motivation from the point of view of random parameter vulnerability.
- The explanation makes sense and the experimental results are strong.

**Reasons To Reject:**

- A key motivation for the paper is that vanilla fine tuning strategy by adding some random parameters can harm the model's performance and robustness . However, some experimental results of the proposed method contradict motivation, making the idea less convincing. By replacing candidates with dummy words such as Jack, John, Ann, the model is still robust. In my opinion, such labels are no better than a random projection layer. How can this avoid hurting the parameters of the model?
In pretrained models, many word embeddings are undertuned or even untuned. For example, some dummy tokens are left for further use. What if you use these words?

- Since the paper highlights the usefulness of prompts for robustness, further experimental study is needed. For example, how do you choose the prompts? How do different prompts affect the robustness of the model? As a baseline, what if you use some dummy prompts, such as "foo bar (mask)"?

- It remains unclear whether robustness arises from the prompt-tuning strategy itself or from some tricky ensemble-style strategy, i.e. by using different models (prompts) with different class logits whose signal may fool the attacker. In Table 2, what is the adversarial accuracy of each prompt? What is the result of "1 prompt + 4 candidates"?

**Reproducibility:**

4: Could mostly reproduce the results, but there may be some variation because of sample variance or minor variations in their interpretation of the protocol or method.

**Reviewer Confidence:**

4: Quite sure. I tried to check the important points carefully. It's unlikely, though conceivable, that I missed something that should affect my ratings.

---

> ### Author Rebuttal · Authors · 2023-08-29
>
> Thank you for your careful assessment of our work and we are happy to see that you found our work to be sound and the proposed method interesting. We would like to emphasize that fine-tuning of language models is an important paradigm in the current landscape of NLP research, where researchers are moving back from “general-purpose” large models, to “task-specific” smaller models (for computational constraints) and/or more accurate models (for instance, even the GPT-3.5 API now supports fine-tuning). Model tuning via prompts is a simple modification that all fine-tuning approaches could easily adopt to enhance robustness.
>
> First, we would like to point out that we found two sources that lead to the robustness difference between MVP (our proposed approach of model-tuning via prompts) and MLP-FT (standard fine-tuning) —
> **first**, as you also note in your review, is the random parameter vulnerability of MLP-FT, and **second**, is the alignment between the pre-training and fine-tuning tasks. We believe the second source explains some of the questions you raised in your review. We now address your concerns below.
>
> ---
>
> > ### Why does using “dummy candidate words” not hurt model robustness, akin to random projection layers?
>
> This is a great and valid concern. Using dummy words like Jack and Ann leads to modifying an embedding of size 768 x C (where C is the number of candidate words) so that they now have a new “meaning”. The effective number of “new parameters” is much lower than the parameters in the “dense 768x768 layer” in the Roberta model. Your suggestion is similar—in terms of parameters to be tuned—to our ablation “Project-CLS” in Figure 3a. As you may see, using ProjectCLS also improves robustness over MLP-FT. This is because we avoid the dense 768x768 layer.
>
> Additionally, **we conducted the new experiment of using empty slots in the vocabulary of Roberta** (which we thought was a great suggestion!) and compared it with using “dummy candidate words” and “class labels”. For the BoolQ dataset, using a Roberta model, we summarize the results below.
>
> Method  | Candidate Choice | Clean Accuracy | Robust Accuracy |
> ----|----|----|----|
> MLP-FT | N/A | 80.6 ± 1.5 | 28.2 ± 1.6
> Project-CLS| N/A | 81.3 ± 0.5 | 37.4 ± 1.2
> MVP | Class Labels         |      82.0 ± 0.6       |     42.9 ± 0.5
> MVP | Dummy Candidate Words (Jack/Ann)      |     80.9 ± 0.3        |    42.1 ± 0.4
> MVP | Empty Slots	  |       81.3 ± 0.3      |      41.2 ± 0.7
>
>
> Based on these accuracies above we find that
> 1. Using class labels is better than using “dummy/untrained words” for both clean and robust accuracy, which supports the random parameter vulnerability hypothesis.
> 2. The robustness achieved upon using completely untrained slots is similar to that when using dummy candidate words. As you also guessed, this suggests that when compared to class labels, modifying dummy words has a similar loss in robustness as with modifying untrained words.
> 3. The MVP models (with random/empty candidate words) are more robust than Project-CLS (which already bridges the robustness gap from MLP-FT). These gains are explained by the pre-training task alignment hypothesis, where pre-training (and fine-tuning) the model with the task of [MASK] infilling helps make the downstream model robust.
>
>
> ---
>
> > ### How are prompts selected, and what is the robustness of different prompts?
>
> We did not perform any sort of prompt tuning for selecting the prompts, and selected prompts primarily from the [OpenPrompt repository](https://github.com/thunlp/OpenPrompt/tree/main/scripts/TextClassification). Further, the choice of prompts used has very little effect on model robustness in the fine-tuning regime. We tabulate the robustness results corresponding to different prompts below (for the BoolQ dataset). Here the first 4 prompts are the prompts we used and "Ruby emerald [MASK]" is similar to your suggestion of "foo bar [MASK]":
>
> Template | Clean Accuracy | Robust Accuracy
> ---- | ---- | ---- |
> Answer to the question is [MASK]        |            81.5 ± 0.5       |      38.5 ± 0.7
> [SEP] [MASK]                                       |            81.7 ± 0.6        |     36.1 ± 0.4
> I think [MASK]                                        |           81.9 ± 0.8        |     35.9 ± 0.2
> [SEP] The answer is [MASK]                 |           82.0 ± 0.3        |     38.1 ± 0.1
> None (MLP-FT) |      80.6 ± 1.5       |       28.2 ± 1.7
> [SEP] Ruby emerald [MASK]                 |            81.4 ± 0.5        |     36.8 ± 0.4
>
>
> ---
>
> > ### How can we be sure that robustness gains are not just from ensembling templates?
>
> We had the exact same concern and for that reason, we conducted an experiment to evaluate the impact of varying the “number of templates”. While adding more templates does increase model robustness, even using a single template is significantly more robust than MLP-FT. One may also wonder, how can we be sure that attacking an MVP model does not get a “false” sense of robustness because it is harder to fool the model this way. To be sure of this, we run experiments on pre-training task alignment where we find that **MVP models _without pre-training_ are actually less robust than MLP-FT**. These results are in Table 3 of the main paper. Hopefully, these findings alleviate your concern.
>
> ---
>
> > ### What is the impact of ensembling the candidates (1 prompt + 4 candidates)
>
> This is not as helpful as ensembling multiple templates, we provide the requested numbers and comparisons below:
>
> Configuration | Clean Accuracy | Robust Accuracy
> ---- | ---- | ---- |
> 1 prompt + 4 candidates |  81.9 ± 0.3        |    37.4 ± 0.5
> 1 prompt + 1 candidate |  81.9 ± 0.8 | 35.9 ± 0.2
> 4 prompt + 1 candidate |  82.0 ± 0.6 | 42.9 ± 0.5
>
> ---
>
> Additionally, you may also be interested in seeing the results of **out-of-distribution robustness** evaluation in [response to Reviewer CusV](https://openreview.net/forum?id=R4yb4m7Nus&noteId=XZi0p9TBKK) which further expands the relevance of our work.
>
> Thank you for your valuable comments. Your questions made us re-think some exciting questions and we believe that the follow-up analysis has further strengthened the work. We look forward to resolving any remaining concerns.

---

### Official Review · Reviewer_nbR2 · 2023-08-03

**Soundness:** 4

**Excitement:**

4: Strong: This paper deepens the understanding of some phenomenon or lowers the barriers to an existing research direction.

**Paper Topic And Main Contributions:**

The paper contributes a novel method to generate a prediction called MVP which is a constrained infilling task (the constrain the tokens we infill to the class categories they are interested in for a particular task) while also attaching a prompt. Further experiments show that MVP is adversarially robust and performing adversarial training on top of this method yields significant improvements in adversarial robustness. The resulting robustness stems from a few design choices (1) removal of the randomly initialized fully connected layer to reduce feature distortion (2) similarity to the pretraining task of Masked Language Modeling.

**Questions For The Authors:**

How much improvement do we expect prompt tuning to give, beyond MVP+Adv?

**Reasons To Accept:**

- Strong experiment results, especially wrt adversarial robustness.
- Human annotators validated efficacy of adversarial example set
- Sound ablation studies and clear analysis that explains the results
- MVP + Adv outperforms MLP-FT + Adv handily

**Reasons To Reject:**

- Only tested on datasets that are simple classification tasks (BoolQ is 2 classes, AGNews contains 4 classes).
- No evaluation for datasets with larger number of classes, no evaluation for non-classification tasks.


**Reproducibility:**

5: Could easily reproduce the results.

**Reviewer Confidence:**

4: Quite sure. I tried to check the important points carefully. It's unlikely, though conceivable, that I missed something that should affect my ratings.

---

> ### Author Rebuttal · Authors · 2023-08-29
>
> Thank you for your positive assessment of our work. We are happy to see that you appreciate the strong empirical gains and simplicity of the proposed approach and the clear analysis and human study. We address your concerns below:
>
> ---
> > *Reasons To Reject:* Only tested on datasets that are simple classification tasks (BoolQ is 2 classes, AGNews contains 4 classes).
>
> We want to clarify that this is _not_ the case. We perform experiments on 5 different datasets. Among these, DBPedia is a 14-class classification problem, and results for the same are presented in Table 5 in the appendix. Secondly, regarding evaluating on more complex datasets, we also perform experiments on MRPC (paraphrase detection) and results for the same are also in Table 5 of the Appendix. Due to space constraints, we were only able to include results for 3 datasets in the main paper. In summary, we perform our experiments on five different datasets—AG News (4-class topic classification), SST2 (binary sentiment classification), BoolQ (boolean question answering), DBPedia14 (14-class topic classification), and MRPC (paraphrase detection). We hope this addresses your concerns.
>
> ---
> > ### How much benefit do you get from Prompt tuning?
>
> To assess the benefit of Prompt tuning, we perform a series of experiments. For instance, we find that using an empty template with just a [MASK] token (which would be at the extreme end of making the prompt weak) also performs very well. These templates are provided in Appendix A. We excluded this discussion in the submission, but we see the value in the same following your comments. We report the model robustness per template chosen for the BoolQ dataset.
>
> Template | Clean Accuracy | Robust Accuracy
> ---- | ---- | ---- |
> Answer to the question is [MASK]        |            81.5 ± 0.5       |      38.5 ± 0.7
> [SEP] [MASK]                                       |            81.7 ± 0.6        |     36.1 ± 0.4
> I think [MASK]                                       |            81.9 ± 0.8        |     35.9 ± 0.2
> [SEP] The answer is [MASK]                 |           82.0 ± 0.3        |     38.1 ± 0.1
> None (MLP-FT) |      80.6 ± 1.5       |       28.2 ± 1.7
>
> So to answer your question: (1) We did not do any prompt tuning for selecting the prompts, and selected them from (or took inspiration from) the [OpenPrompt repository](https://github.com/thunlp/OpenPrompt/tree/main/scripts/TextClassification) wherever available. (2) The choice of prompts used has a small (2%) effect on model robustness in the fine-tuning regime (unlike few shot or in-context learning regimes). (3) We believe that other prompt tuning methods (eg. https://aclanthology.org/2022.acl-long.158/) may yield further improvements. Benchmarking them is an interesting future direction that is outside the scope of this work.
>
> ---
>
> We hope this further strengthens your confidence in MVP. Additionally, you may also be interested in seeing the results of **out-of-distribution robustness** evaluation in [response to Reviewer CusV](https://openreview.net/forum?id=R4yb4m7Nus&noteId=XZi0p9TBKK) which further expands the relevance of our work.

---

### Official Review · Reviewer_CusV · 2023-08-07

**Soundness:** 4

**Excitement:**

3: Ambivalent: It has merits (e.g., it reports state-of-the-art results, the idea is nice), but there are key weaknesses (e.g., it describes incremental work), and it can significantly benefit from another round of revision. However, I won't object to accepting it if my co-reviewers champion it.

**Paper Topic And Main Contributions:**

This work explores the adversarial robustness of model-tuning via prompts (MVP) and compares it to models which are fine-tuned.  MVP here is not an efficient tuning method and all the weights of the model are trained during fine-tuning on prompts. During inference they ensemble the results from different prompts. The authors demonstrate that their method has better robustness under a number of perturbation methods on 5 classification datasets. Moreover they demonstrate sample efficiency over fine-tuning.

**Questions For The Authors:**

1. How does method scale to different classification tasks. How difficult would designing the prompts be ?

2. Did you discard any prompts which did not perform well ?

**Reasons To Accept:**

1. Using prompts to avoid adding random parameters for adversarial robustness is a nice approach and it works on the datasets presented in the paper. MVP shows better adversarial robustness compared to fine-tuning without sacrificing on accuracy.

2. The experiments confirm the issue with adding random features in the fine-tuning stage as raised by Kumar e. al 2022 etc. fine-tuning using prompts is an alternate to the LPFT.

**Reasons To Reject:**

1. Lack of discussion and analysis on the inference latency as more templates are added for MVP.

2. This paper would have benefited from a direct comparison between LPFT and MVP on both OOD generalization and Adversarial robustness. Both these approaches combat the "brittleness" of the fine-tuning process. It would have been interesting to see LPFT vs MVP using different number of prompts. A discussion on the relative advantages of each approach would have added to this work.


**Reproducibility:**

4: Could mostly reproduce the results, but there may be some variation because of sample variance or minor variations in their interpretation of the protocol or method.

**Reviewer Confidence:**

4: Quite sure. I tried to check the important points carefully. It's unlikely, though conceivable, that I missed something that should affect my ratings.

---

> ### Author Rebuttal · Authors · 2023-08-29
>
> Thank you for your careful assessment of our work and thought-provoking questions. We are happy to see that you found our method to be sound which led to strong gains accompanied by a meaningful analysis. We address your concerns below.
>
> ---
>
>
> > ### What is the inference latency of using multiple templates?
>
> We present the latency numbers below and also compare them with the latency of the standard MLP-FT approach. In particular, we measure the time required for 2000 forward passes of data from the IMDb dataset with a batch size of 1. T= 24.45 ± 0.25 secs.
>
> Configuration | Time |
> ---- | ---- |
> MLP-FT | 23.25 ± 0.37 secs (0.95*T)
> 1 Template |   24.45 ± 0.25 secs (T)
> 2 Templates|  27.51 ± 0.45 secs (1.15 * T)
> 3 Templates|  32.81 ± 0.30 secs (1.34 * T)
> 4 Templates| 35.65 ± 0.44 secs (1.45 * T)
>
> In summary, using multiple templates makes predictions about 1.45x slower, but leads to improved robustness. We will include these results in the draft.
>
> ---
> > ### Discussion on OOD robustness:
>
> We performed **new experiments** to assess the out-of-distribution robustness of MVP, MLP-FT, and LPFT based on your feedback, and present results below. We use 5 sentiment classification datasets, namely SST2, Amazon Polarity, IMDb, Movie Rationales, and Rotten Tomatoes. We fine-tune a Roberta model on 1000 examples of each of these datasets and evaluate on every other dataset. The full table of results is provided below, but in summary, we see that across 5 different models (and 20 evaluations) the average accuracy for MVP (89.65%) is 2% more than MLP-FT and 1.3% more than that of LP-FT.
>
> |  | | SST2 | | | | | Amazon | Polarity | | | | IMDb | | | | | Movie | Rationales | | | | Rotten | Tomatoes |
> |------------------|----------------|--------------|-------------|-------------------|-------------------|-------------------------|--------------|-------------|-------------------|-------------------|-----------------|--------------|-------------|-------------------|-------------------|---------------------|--------------|-------------|-------------------|-------------------|---------------------|--------------|-------------|
> | **Train\\Eval** | **MVP** | **MLP-FT** | **LPFT** | | | **MVP** | **MLP-FT** | **LPFT** | | | **MVP** | **MLP-FT** | **LPFT** | | | **MVP** | **MLP-FT** | **LPFT** | | | **MVP** | **MLP-FT** | **LPFT** |
> | SST2 | 91.31 | 91.2 | 91.9 | | | 92.8 | 89.21 | 90.1 | | | 89.45 | 87.65 | 87.9 | | | 86 | 85.93 | 86.23 | | | 86.12 | 83.21 | 84.09 |
> | Amazon Polarity | 90.86 | 88.53 | 88.98 | | | 92.9 | 92.9 | 93.4 | | | 92 | 91.2 | 91.1 | | | 85.92 | 83.92 | 84.23 | | | 86.12 | 83.3 | 84.5 |
> | IMDb | 84.4 | 81.42 | 83.5 | | | 91.85 | 88.8 | 88.7 | | | 92.2 | 91.85 | 92.4 | | | 91.95 | 89.95 | 90.15 | | | 81.92 | 78.14 | 80.12 |
> | Movie Rationales | 89.91 | 85.89 | 85.45 | | | 92.5 | 89.05 | 90.7 | | | 91.7 | 90.6 | 91.6 | | | 94.4 | 93.46 | 94.3 | | | 87.36 | 83.02 | 83.43 |
> | Rotten Tomatoes | 92.43 | 92.08 | 92.9 | | | 92.6 | 89.5 | 90.4 | | | 90.9 | 88.55 | 90.2 | | | 86.45 | 83.92 | 84.67 | | | 87.15 | 87.05 | 87.2 |
> | **Average** | 89.782 | 87.824 | 88.546 | | | 92.53 | 89.892 | 90.66 | | | 91.25 | 89.97 | 90.64 | | | 88.944 | 87.436 | 87.916 | | | 85.734 | 82.944 | 83.868 |
>
> These results show that MVP is superior to MLP-FT and LPFT for both adversarial and OOD robustness. In summary, LPFT helps reduce the impact of random parameter vulnerability, but MVP additionally allows pre-training task alignment (the second hypothesis) hence resulting in superior performance and no fundamental trade-off be it OOD or adversarial robustness.
>
>
> ---
> > ### How does the method scale to different classification tasks?
>
> In our work, we evaluate the effectiveness of MVP on 5 different datasets–AG News (4-class topic classification), SST2 (binary sentiment classification), BoolQ (boolean question answering), DBPedia14 (14-class topic classification), and MRPC (paraphrase detection). For each of these tasks, we made no particular adaptations to make the method work. Therefore, there is almost no cost to setting up a new task, both in terms of complexity (like the method scales to tasks with a large number of labels like 14 in DBpedia) and style (like paraphrase detection in the case of MRPC).
>
> > ### How are prompts selected for new tasks?
>
> (1) We do not do any prompt tuning to make MVP work and do not discard any prompts. For the most part, we selected prompts from (or took inspiration from) the [OpenPrompt repo](https://github.com/thunlp/OpenPrompt/tree/main/scripts/TextClassification) whenever the corresponding datasets were available. (2) The choice of prompts used has very little effect on model robustness in the fine-tuning regime. For instance, we find that using an empty template with just a [MASK] token (which would be at the extreme end of making the prompt weak) also performs very well. These templates are provided in Appendix A and detailed per-prompt accuracies may be found in response to Reviewer QuR4.
>
> ---
> Once again, thank you for these pointers that have further strengthened our work. We will include our new results in the final draft and look forward to clarifying any remaining concerns.

---

### Meta-Review · Area_Chair_bAPs · 2023-09-19

**Recommendation:** 4

**Metareview:**

The paper introduces Model-tuning via Prompts (MVP) as a novel approach to enhance adversarial robustness in NLP models. MVP involves fine-tuning a model on prompts and ensemble results during inference, demonstrating superior robustness without accuracy loss on various datasets. Reviewers found the idea is interesting and the experiments are strong. They also raise concerns about limited evaluations on simple tasks, the need for a comparison with LPFT, and the scalability of MVP. Overall, this paper is of high quality and I therefore recommend acceptance.

---

### Decision · Program_Chairs · 2023-10-07

**Decision:**

Accept-Main

**Comment:**

The paper introduces Model-tuning via Prompts (MVP) as a novel approach to enhance adversarial robustness in NLP models. MVP involves fine-tuning a model on prompts and ensemble results during inference, demonstrating superior robustness without accuracy loss on various datasets. Reviewers found the idea is interesting and the experiments are strong. They also raise concerns about limited evaluations on simple tasks, the need for a comparison with LPFT, and the scalability of MVP. Overall, this paper is of high quality and I therefore recommend acceptance.